# All-polymer organic solar cells with nano-to-micron hierarchical morphology and large light receiving angle

Rui Zeng[1,9], Lei Zhu [1,9], Ming Zhang [1], Wenkai Zhong [1], Guanqing Zhou[1], Jiaxing Zhuang[1], Tianyu Hao[1], Zichun Zhou[1], Libo Zhou[1], Nicolai Hartmann [2], Xiaonan Xue[3], Hao Jing[3], Fei Han[4], Yiming Bai[4], Hongbo Wu[5], Zheng Tang [5], Yecheng Zou[6], Haiming Zhu [7], Chun-Chao Chen [8], Yongming Zhang[1] & Feng Liu [1] ✉

Distributed photovoltaics in living environment harvest the sunlight in different incident angles throughout the day. The development of planer solar cells with large light-receiving angle can reduce the requirements in installation form factor and is therefore urgently required. Here, thin film organic photovoltaics with nano-sized phase separation integrated in micro-sized surface topology is demonstrated as an ideal solution to proposed applications. All-polymer solar cells, by means of a newly developed sequential processing, show large magnitude hierarchical morphology with facilitated exciton-to-carrier conversion. The nano fibrilar donor-acceptor network and micron-scale optical field trapping structure in combination contributes to an efficiency of 19.06% (certified 18.59%), which is the highest value to date for all-polymer solar cells. Furthermore, the micron-sized surface topology also contributes to a large light-receiving angle. A 30% improvement of power gain is achieved for the hierarchical morphology comparing to the flat-morphology devices. These inspiring results show that all-polymer solar cell with hierarchical features are particularly suitable for the commercial applications of distributed photovoltaics due to its low installation requirement.

Single-junction organic solar cells (OSCs) have achieved prominent power conversion efficiencies (PCEs) over 19% in recent years[1–4]. The recent development of polymerized non-fullerene acceptors (NFAs) that link NFAs together with π-spacers shows greatly improved stability, making all-polymer solar cells (APSCs) more favorable for

commercial applications[5–9]. Nevertheless, the performance of APSCs remains low compared to the NFA-based solar cells, which is due to the difficulty in obtaining a suitable morphology for the studied systems. In fact, the rigid chain of polymeric donor and acceptor mixture often induces strong polymer crystallization and phase segregation,

[1]School of Chemistry and Chemical Engineering, Frontiers Science Center for Transformative Molecules, In-situ Center for Physical Science, and Center of Hydrogen Science Shanghai Jiao Tong University, Shanghai 200240, China. [2]Attocube Systems AG, Eglfinger Weg 2, Haar 85540, Germany. [3]Shanghai OPV Solar New Energy Technology Co., Ltd, Shanghai 201210, China. [4]State Key Laboratory of Alternate Electrical Power System with Renewable Energy Sources, North China Electric Power University, Beijing 102206, China. [5]Center for Advanced Low-dimension Materials, State Key Laboratory for Modification of Chemical Fibers and Polymer Materials, College of Materials Science and Engineering Donghua University, Shanghai 201620, China. [6]State Key Laboratory of Fluorinated Functional Membrane Materials and Dongyue Future Hydrogen Energy Materials Company, Zibo City, Shandong 256401, China. [7]Department of Chemistry, Zhejiang University, Hangzhou Zhejiang 310027, China. [8]School of Materials Science and Engineering, Shanghai Jiao Tong University, Shanghai 200240, China. [9]These authors contributed equally: Rui Zeng, Lei Zhu. ✉e-mail: fengliu82@sjtu.edu.cn

resulting in a bicontinuous network with low domain purity, which is detrimental to carrier transport. In addition, the application of solar cells in building structure intergradation requires more flexible form factor and large light-receiving angle tolerance to ensure a more constant power output upon daytime solar radiation angle change. Thus, manipulating the thin film morphology of APSCs from nano to micron length scales to enable strong light trapping, high carrier generation efficiency, and fast carrier transport is critical for efficiency improvement as well as light-receiving angle tolerance.

In order to address the above-mentioned challenges, we developed a consequential processing methodology to induce nano-to-micron sized hierarchical morphology based on the state-of-the-art PM6:PY-IT all-polymer blends[10–12]. The critical knack is to synergize the effect from the phase-transitional solid additive of 1,4-diiodobenzene (DIB)[2,13], thermal annealing and solvent vapor annealing (SVA)[14,15] post-treatments. A more effective control on the conjugated polymer crystallization is achieved through the controlled removal of solvent and solid additive, by which the polymer crystallization kinetics and thin film surface structure can be manipulated. A high crystallinity interconnected double-fibril network morphology is obtained, showing superiority in generating high exciton splitting and high carrier transport[14]. And the DIB diffusion and removal onto thin film top surface leads to difference in modulus and tension, making mechanical instability between the surface layer and the bulk[16–18]. Surface wrinkle pattern is created, and the nano-sized fibril network is obtained and integrated within the bulk and surface. In the subsequent SVA treatment, the swelling and re-drying of the blended thin film cause shrinkage[19], leading to the change of wrinkle pattern from ridge to island. This morphology is an important milestone comparing to the previous nano-sized morphology optimization, in which such morphology can induce favorable optical trapping for photoactive thin films. The combinate nanoscale phase separation and micro surface pattern leads to an unprecedented high PCE of 19.06% (certified 18.59%) for APSCs. A large solar light receiving angle of unidirectional 50° is achieved (85% of the 0° PCE). These advances endow easier intergradation of OSC devices for commercial application scenarios. The power output at summer solstice in Shanghai city is calculated, which shows a ~30% improvement. The development of the surface-to-bulk hierarchical morphology can be as important as to the surface pattern for silicon solar cells. The high efficiency, high stability, and high-power output visualize the high potential commercial application for APSCs.

## Results

### The significance of surface pattern

The surface pattern fabrication is a milestone for silicon solar cells[20,21], which serves as light trapping structure to suppress the optical loss and increase the optical path[22,23]. Dating back to 1986 at the University of New South Wales, surface optical structure has become the key technology for silicon cells that enables efficiency breakthrough over 20%[24]. And subsequently, silicon solar cells experienced vigorous developments. Developing surface structure for OSC can be more difficult due to the flow levering of solution processing methods. We have tried thin film buckling of P3HT:PC$_{71}$BM blends to induce surface patterns by using the different mechanical strength of the OPV thin film and top electrodes[25,26] (relevant data is shown in Supplementary Fig. 1). However, this methodology is not compatible for the currently used high efficiency systems. Surface patterning methods such as photolithography and wet etching can be considered[27,28], which, however, induces much more complex device fabrication procedure. Thus, implementing surface pattern with a suitable morphology for OPV thin film is a challenging task. To obtain a rugged surface pattern to better utilize solar radiation, a unique surface formation mechanism needs to be developed. Here, surface pattern is achieved by using a phase transformative additive, which can crystallize and vaporize on a

semi-dried thin film. This idea is developed into a consequential processing method to induce hierarchical morphology, which not only optimizes the nano-sized fiber crystallites but also induces micron-sized surface optical morphology. Figure 1 shows the idea of optical structure in silicon solar cells and current efficient non-fullerene OSCs. The significant role of enhanced light absorption and light-receiving angle broadening originated from surface optical structure is expected to significantly improve the power output in daytime solar cell applications[22].

### Thin film fabrication and device performance

The optical properties of the flat and hierarchical thin films are compared. Figure 2a and Supplementary Fig. 2 show the chemical structures and absorption features of PM6 and PY-IT. Chloroform (CF) is used as the fast-drying solvent while DIB is used as the phase transformative additive due to its easy sublimation nature. Carbon disulfide (CS$_2$) solvent vapor annealing (SVA) is used to induce the thin film swelling. Thermal annealing (TA) is used to optimize morphology and synchronously vaporize DIB. These processing are combined in different sequences to manipulate the thin film morphology. The absorption of PY-IT in chloroform (CF) shows red-shift after adding DIB. As a result, a 12 nm red-shift can be seen when the DIB concentration is 30 mg mL$^{-1}$, suggesting the planarization of molecular conformation of PY-IT under the presence of DIB. This feature could induce better ordering of PY-IT during film formation[29]. The TA and SVA treatments is used to refine the thin-film crystallization and bicontinuous network formation. The photoluminescence (PL) quenching experiment (Supplementary Fig. 3) was used to access the phase separation of blended thin films. It is seen that DIB and DIB/TA/SVA processed thin films showed enhanced PL quenching efficiencies, which, in conjugation with enhanced crystallization, indicate a more efficient exciton diffusion for photoactive blends. A better-balanced mixing and phase separation condition is obtained in an optimized bicontinuous nano-sized morphology that supports the highly efficient exciton diffusion and dissociation.

We then investigate the morphology formation mechanism by comparing DIB with a conventionally used 1-chloronaphthalene (CN) additive. The thin film drying kinetics were shown in Supplementary Fig. 4. It is seen that CF solvent vaporizes within seconds which is not beneficial to the crystallization process of donor and acceptor materials. On the other hand, CN additive dramatically prolongs the film drying, leaving longer time for molecular reorganization. However, DIB additive behaves differently. For the case of DIB, the solvent vaporizes quickly as that in CF processing to form a thin film. Then, a slow and steady DIB removal process is recorded, which is due to DIB diffusion and sublimation. The film thickness right after casting and post treatments were compared (shown in Supplementary Fig. 5). The DIB processed thin film showed an initial thickness around 200 nm. After TA treatment at 100 °C, the film thickness was quickly reduced to about 120 nm in seconds, which is close to natural film thickness processed without additive. The SVA treatment of DIB processed thin film further induced large sized DIB crystallization on surface, which could not be removed by TA. Thus, a controlled removal of DIB is critical. The detailed fabrication DIB/TA/SVA process is shown in Supplementary Fig. 6, and more details are summarized in Device Fabrication section.

The J-V curves and solar cell performances for the PM6:PY-IT blends are summarized in Fig. 2b and Table 1. The device optimization process is shown in Supplementary Fig. 7 and Supplementary Table 1. Supplementary Fig. 8 shows the energy levels of PM6 and PY-IT. The lowest unoccupied molecular orbital (LUMO) and highest occupied molecular orbital (HOMO) energy levels were estimated by cyclic voltammetry (CV). The LUMO/HOMO levels are −3.67/−5.60 eV for PM6 and −3.88/−5.71 eV for PY-IT. Thus, a heterojunction with a high built-in potential is expected (LUMO$_{PY-IT}$-HOMO$_{PM6}$). The as-

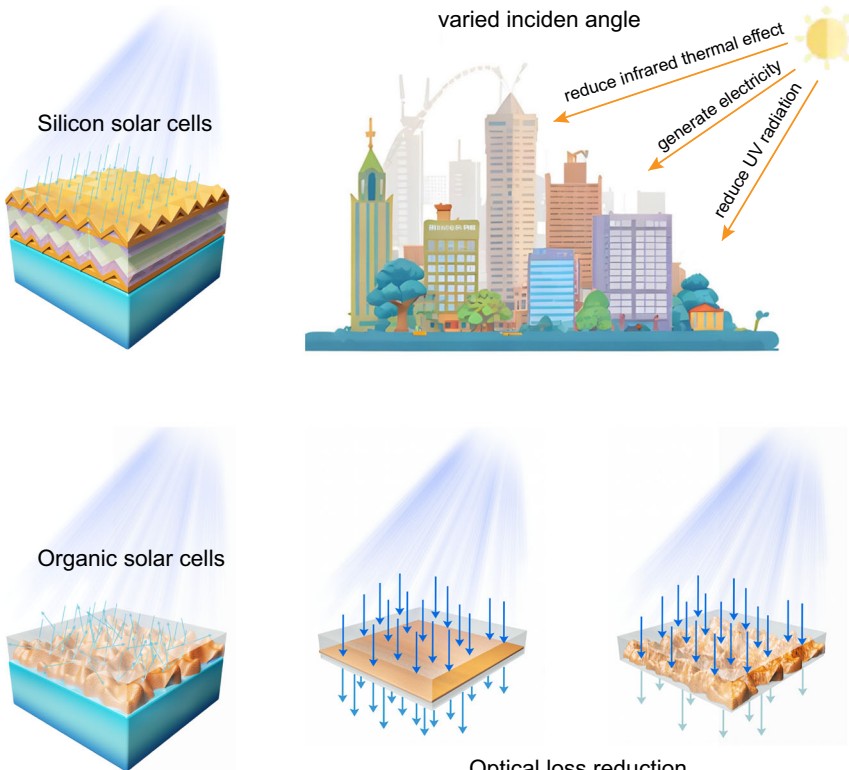

Silicon solar cells

varied inciden angle

reduce infrared thermal effect

generate electricity

reduce UV radiation

Organic solar cells

Optical loss reduction

**Fig. 1 | The past and present of surface topology and its significance on device function.** The comparison of surface topography between silicon solar cells and organic solar cells is shown in the left two figures, and the lower right image shows that the surface topography reduces light loss in organic solar cells. The figure on the top right indicates that organic solar cells with such surface topography are more suitable for urban environments.

cast, DIB and DIB/TA/SVA processed blends showed a PCE of 15.17% ($V_{OC}$ = 0.933 V, $J_{SC}$ = 24.12 mA cm$^{-2}$, FF = 67.42%), 16.97% ($V_{OC}$ = 0.938 V, $J_{SC}$ = 24.98 mA cm$^{-2}$, FF = 72.39%) and 18.32% ($V_{OC}$ = 0.944 V, $J_{SC}$ = 25.92 mA cm$^{-2}$, FF = 74.88%), respectively. Moreover, unprecedented high efficiency of 19.06% is obtained with a $V_{OC}$ of 0.945 V, a $J_{SC}$ of 26.37 mA cm$^{-2}$, and an FF of 76.48%, when the 2PACz was introduced as hole transport layer (HTL). Supplementary Fig. 9 shows the $J$-$V$ curves. The optimum device obtained a third-party certified PCE of 18.59% (Supplementary Figs. 10–13), which is the highest value for APSCs. The higher $V_{OC}$ and $J_{SC}$ in DIB/TA/SVA processed thin film indicate that both carrier generation and carrier transport are optimized, suggesting that a more favorable morphology is obtained. Supplementary Fig. 14 shows the normalized performance of encapsulated device under continuous illumination over 1200 h under white light illumination of 1 sun intensity in air. The DIB/TA/SVA device efficiency stabilizes at around 800 h, which is ahead of DIB (around 950 h) and as-cast (1100 h) devices. The steady-state efficiency of DIB/TA/SVA device remained at 80% compared with 74% for DIB device and 68% for as-cast devices. These results imply that the morphology of DIB/TA/SVA device is more stable. The external quantum efficiencies (EQEs) and differences are shown in Fig. 2c and Supplementary Fig. 15. It is seen that DIB/TA/SVA processing improved EQE value by ~4% with a wide wavelength range (from 350 nm to 840 nm), which originates from the improved exciton dissociation and carrier collection. As for DIB/SVA/TA condition, high concentration of DIB accumulates on the surface of the active layer, which reduces carrier extraction and results in lower EQE (Supplementary Fig. 16). The internal quantum efficiency (IQE) and EQE are further simulated electro-optically based on optical transfer matrix and drift diffusion model. The optical constants of refractive-index ($n$) and extinction-coefficient ($k$) for the blends and all ancillary layers are shown in Supplementary Fig. 17. The comparison of

simulated results and measured value are summarized in Supplementary Fig. 18a. The solid line represents experimental results, while the dash line is simulation results. The simulation results are in well agreement with the test results, which validates the highest EQE/IQE value of DIB/TA/SVA processed device, and confirms the elevation of $J_{SC}$. We further simulate the changes of EQE as a function of junction thickness (Supplementary Fig. 18b). The shape of EQE profiles becomes flat when the film thickens increases. For the active film of 120 nm, most of the photons at absorption dip around 660 nm can be utilized, which contributes to improved $J$sc and agrees well with the measured result. These results set the basis for thickness optimization, and explain the fundamental reason of the differences between absorption and EQE profiles. The statistical PCE for different device preparation conditions is summarized in Fig. 2d, which demonstrates the good reproducibility of the device performance and high reliability of the processing strategy. An explicit comparison to the state-of-the-art APSC performance is given in Fig. 2e and Supplementary Fig. 19 based on the current results and 95 data points from 69 literatures, from which we see that the newly develop APSC blends and morphology enable best balanced $J_{SC}$ and $V_{OC}$.

## Nano-to-micron morphology characterization

We investigated the nanoscale morphology of blended thin films. Grazing incidence wide-angle X-ray scattering (GIWAXS) was performed to study the polymer crystallization[30]. The scattering patterns and sector averaged curves in the in-plane (IP) and out-of-plane (OOP) directions are summarized in Fig. 3b and Supplementary Figs. 20–21. Figure 3c underlined the difference of crystallization behavior under different processing methods. Comparing with the as-cast PM6:PY-IT blended film, crystallization improvements can be seen in the DIB and DIB/TA/SVA processed blends, since the ~0.28 Å$^{-1}$ and the ~1.67 Å$^{-1}$ peak intensity increases significantly. The crystalline coherence

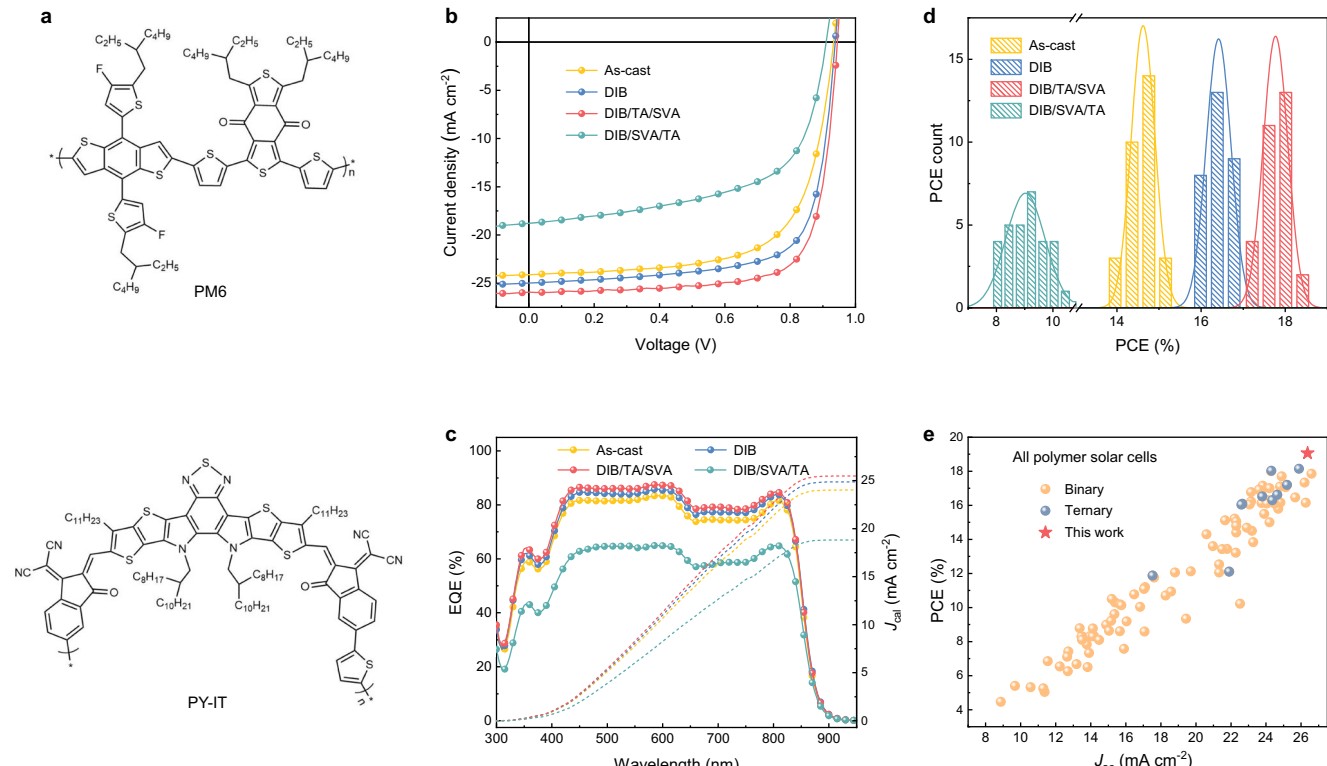

**Fig. 2 | Materials and devices performances. a** Chemical structures of PM6 and PY-IT. **b** J-V curves of the APSCs in different condition under AM 1.5 G, 100 mA cm⁻². **c** EQE spectra of the APSCs in corresponding condition. All the DIB condition in text is followed by a TA process. **d** Histogram of PCE measurement for devices of the APSCs in corresponding condition under AM 1.5 G, 100 mA cm⁻². **e** Plots of the PCE versus $J_{SC}$ for the efficient all-PSCs reported in the literature.

### Table 1 | Photovoltaic performance under simulated AM 1.5 G illumination

| PM6:PY-IT | $J_{SC}$ (mA cm⁻²) | $J_{Cal}$ (mA cm⁻²) | $V_{OC}$ (V) | FF (%) | PCE[a] (%) |
|---|---|---|---|---|---|
| As-cast | 24.12 (23.74 ± 0.18) | 24.03 (23.79 ± 0.14) | 0.933 (0.929 ± 0.003) | 67.42 (66.26 ± 0.79) | 15.17 (14.62 ± 0.28) |
| DIB | 24.98 (24.41 ± 0.18) | 24.87 (24.62 ± 0.13) | 0.938 (0.937 ± 0.002) | 72.39 (71.63 ± 0.85) | 16.97 (16.42 ± 0.30) |
| DIB/TA/SVA | 25.92 (25.38 ± 0.25) | 25.48 (25.21 ± 0.14) | 0.944 (0.941 ± 0.002) | 74.88 (74.40 ± 0.49) | 18.32 (17.76 ± 0.29) |
| DIB/SVA/TA | 18.79 (18.08 ± 0.87) | 18.82 (18.59 ± 0.17) | 0.910 (0.905 ± 0.009) | 59.64 (55.10 ± 2.86) | 10.20 (9.03 ± 0.69) |
| DIB/TA/SVA[b] | 26.37 (25.96 ± 0.20) | | 0.945 (0.940 ± 0.003) | 76.48 (76.10 ± 0.48) | 19.06 (18.56 ± 0.18) |
| Certified | 26.26 | | 0.945 | 74.95 | 18.59 |

[a]The average values are obtained from repeatedly individual experimental results.
[b]The device prepared with 2PACz as HTL.

lengths (CCLs) fitted for the lamellar stacking peaks (Supplementary Tables 2–3) are 58, 70, 74 Å for as-cast, DIB, and DIB/TA/SVA processed blends, respectively. As for π-π stacking peaks, the CCLs are 17, 18, 19 Å for as-cast, DIB, and DIB/TA/SVA blends, respectively. Pronounced increase is seen for DIB/TA/SVA blends, indicating that the consequential processing enhances the ordering of π-π stacking. Such change mostly arises from TA/SVA treatment, which drives out DIB additive and induces fibrillar morphology formation. The improved π-π stacking is beneficial to charge transport and recombination properties, thus better device performance is justified.

The phase separation of the blended thin films was investigated by atomic force microscopy (AFM) and transmission electron microscopy (TEM) (Supplementary Fig. 22). The as-cast film shows a homogeneous mixing morphology. A fibrillar-type morphology is seen for DIB and DIB/TA/SVA blends, suggesting that the enhanced crystallization assists the growth of polymer fibrils. Atomic force microscopy-based infrared spectroscopy (AFM-IR) was used to probe the fibril structure in the DIB/TA/SVA blends[31,32], using the specific infrared (IR)

absorption at 1650 and 2215 cm⁻¹ for PM6 and PY-IT, respectively (Supplementary Fig. 23). As shown in Fig. 3a, PM6 and PY-IT show well-resolved interconnected fibrillar structures with an averaged width of ~15 nm. The inter-fibril distance is of ~30 nm, as estimated by the line-cut profiles. The carbon and nitrogen K-edge resonant soft X-ray scattering (CK-RSoXS and NK-RSoXS) were performed at 285.6 and 399.8 eV to study the phase separation statistics (Supplementary Fig. 24)[33,34]. At 285.6 eV, scattering contrast rising from C 1s → π* transitions delocalized over the π-conjugated polymer backbones. For DIB and DIB/TA/SVA blends, scattering shoulders are seen at ~0.014 and ~0.017 Å⁻¹, corresponding to phase separation of 45 and 37 nm. At 399.8 eV, the scattering contrast stems from the N 1s → π* transitions on PY-IT. The DIB and DIB/TA/SVA blends show weak shoulders at ~0.013 and ~0.017 Å⁻¹, corresponding to the PY-IT inter-fibril distances of 48 and 37 nm, respectively. Thus, a refined fibrillar morphology is obtained upon DIB/TA/SVA processing. This morphology is key in elevating the photophysical process of the photoactive layer to deliver higher $J_{SC}$ and FF.

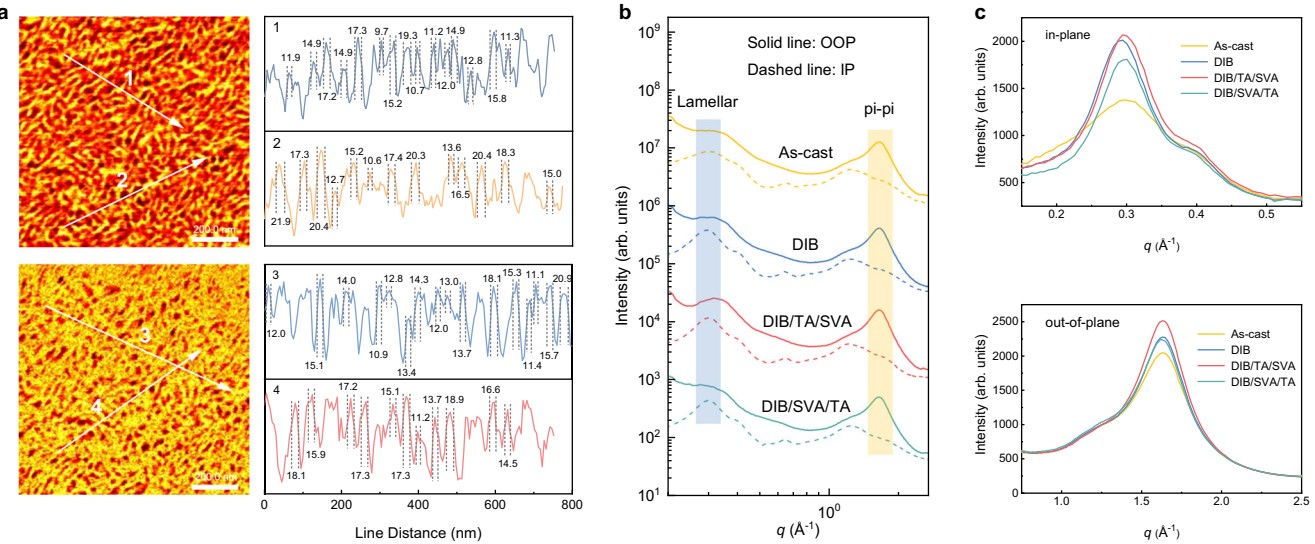

**Fig. 3 | Nanoscale fibril crystallization morphology characteristics. a** Tapping AFM-IR image at the wavenumber of 1650 cm$^{-1}$ and 2215 cm$^{-1}$ in optimal DIB/TA/SVA condition. **b** Line cut profiles for PM6:PY-IT blends in different conditions. **c** Individual in plane and out of plane line cut profiles of the 2D GIWAXS data.

On top of the nano-sized bicontinuous morphology, a micron-sized topology is seen for sequentially processed thin films. Optical microscope, surface profiler, and AFM were used to reveal the structure details. As shown in Fig. 4a–c and Supplementary Figs. 25–26, DIB processed thin film displays a uniform surface over hundreds of micrometers. TA treatment leads to ridge structure in tens of microns. After subsequent SVA treatment, island microstructure is formed. The TA and SVA processing sequence cannot be changed since the controlled removal of DIB is key. SVA on DIB processed thin film results in large sized DIB spherical crystals and covers the entire film surface (Supplementary Fig. 26e–f). The DIB/TA/SVA surface features are clearly visualized by surface profilometer and AFM. The island structure affords diffusive reflection to trap light and improve device performance from back as that in silicon solar cells in front[35]. We measured the angle dependent device performance of solar cells (Fig. 4d–f and Supplementary Table 4). For DIB/TA/SVA device, a 16.76% PCE is obtained at an incident angle of 50°. For comparison, the flat device (as-cast) showed a 11.02% PCE at the same angle of incidence. In this scenario, higher daytime total energy production and more flexible installation geometry can be obtained. The developed sequential processing was extended to a few commonly used OSC blends (Fig. 4e, Supplementary Figs. 27–28, and Supplementary Tables 5–8), encouraging results were obtained. Thus DIB/TA/SVA processed can be a widely used strategy to induce nano-to-micron hierarchical morphology to enhance light receiving angle responses.

### Device physics and photophysical processes

To reveal the function of surface pattern, we performed the optical electric field $|E(x)^2|$ distribution simulation using transfer matrix method and measured $n$ and $k$ as shown in Supplementary Fig. 17 to evaluate the impact of surface feature on light field distribution in photoactive thin film under varied light incident angles. Supplementary Fig. 29a–f show the results of flat device (as-cast), and Supplementary Fig. 29g–l show the results of rugged device (DIB/TA/SVA). It is seen that even at the very small light incidence angle (for example 0° deviation angle in Supplementary Fig. 29a, g) the device with rugged surface shows less optical loss due to light trapping effect. This trend is remained for all light incidence angles. Thus, the sequential DIB/TA/SVA processed device shows less optical loss and larger $J_{SC}$. The 2D volume integration of light fields is summarized in Supplementary Table 9. The detailed value is compared with measured $J_{SC}$ in Supplementary Fig. 30. The solid blue line and red line represent the $J_{SC}$ for as-

cast and DIB/TA/SVA condition at different incident angles, respectively. The dash blue line and red line represent the 2D integrate value of light field distribution for as-cast and DIB/TA/SVA condition at different incident angles, respectively. The simulation results are in good agreement with the experimental value, revealing the advantage of the rugged surface feature of DIB/TA/SVA processed device. The impact of light incidence angle on power generation is estimated using the summer solstice of Shanghai city. The power gain is calculated using the solar radiation data at device operation conditions (Fig. 4g). The changes of zenith angle ($\theta_Z$) during summer solstice is shown in Supplementary Table 10. The calculated the power output of the devices is shown in Fig. 4h, Supplementary Fig. 31, and Supplementary Table 11. It is seen that the power generation for DIB/TA/SVA processed device is increased by ~30%, which is a big leap for solar cells in application. It should also be noted that the improved light incidence angle response for DIB/TA/SVA processed device is beneficial for varied installation conditions (flat, tilted, or even vertical), which solves the form factor limitation in building integrated applications.

The synergy of nano fibril network and rugged surface pattern contributes to the improved photon-to-carrier conversion. The spectral and temporal characteristics of the charge-transfer dynamics in blended thin films are studied using the transient absorption spectroscopy (TAS)[36,37]. The exciton diffusion lengths ($L_D$s) for as-cast, DIB, DIB/TA/SVA processed neat PY-IT films are 35 nm, 38 nm, and 40 nm, respectively (Supplementary Fig. 32 and Supplementary Table 12). Thus, PY-IT has an intrinsic high $L_D$ that facilitates light extraction. An 800 nm excitation is used to excite PY-IT only, which is the major avenue of carrier generation. The 2D spectrum and representative TAS profiles at indicated delay times are shown in Supplementary Figs. 33–34. The decay traces at ~850 nm represent the ground-state bleach (GSB) of PY-IT, and the decay traces at around 910 nm are assigned to excited states absorption (ESA) of PY-IT. With the decay of the PY-IT bleach peak (850 nm), the PM6 GSB peak at around 595 nm rises, suggesting the hole-transfer process from PY-IT to PM6. These profiles are summarized in Fig. 5a, and fitted by biexponential function to obtain the kinetics (Supplementary Table 13). The fast components ($\tau_1$) represent the kinetics of the exciton dissociation in mixing domain or at interfaces, which are 0.53, 0.45 and 0.42 ps for the as-cast, DIB, and DIB/TA/SVA processed blends, respectively. The slow components ($\tau_2$) represent the kinetics of exciton diffusion in crystalline domain to interfaces, which are 13.71, 12.07 and 11.37 ps for the as-cast, DIB, and DIB/TA/SVA processed blends, respectively. The reduced $\tau_1$ indicates

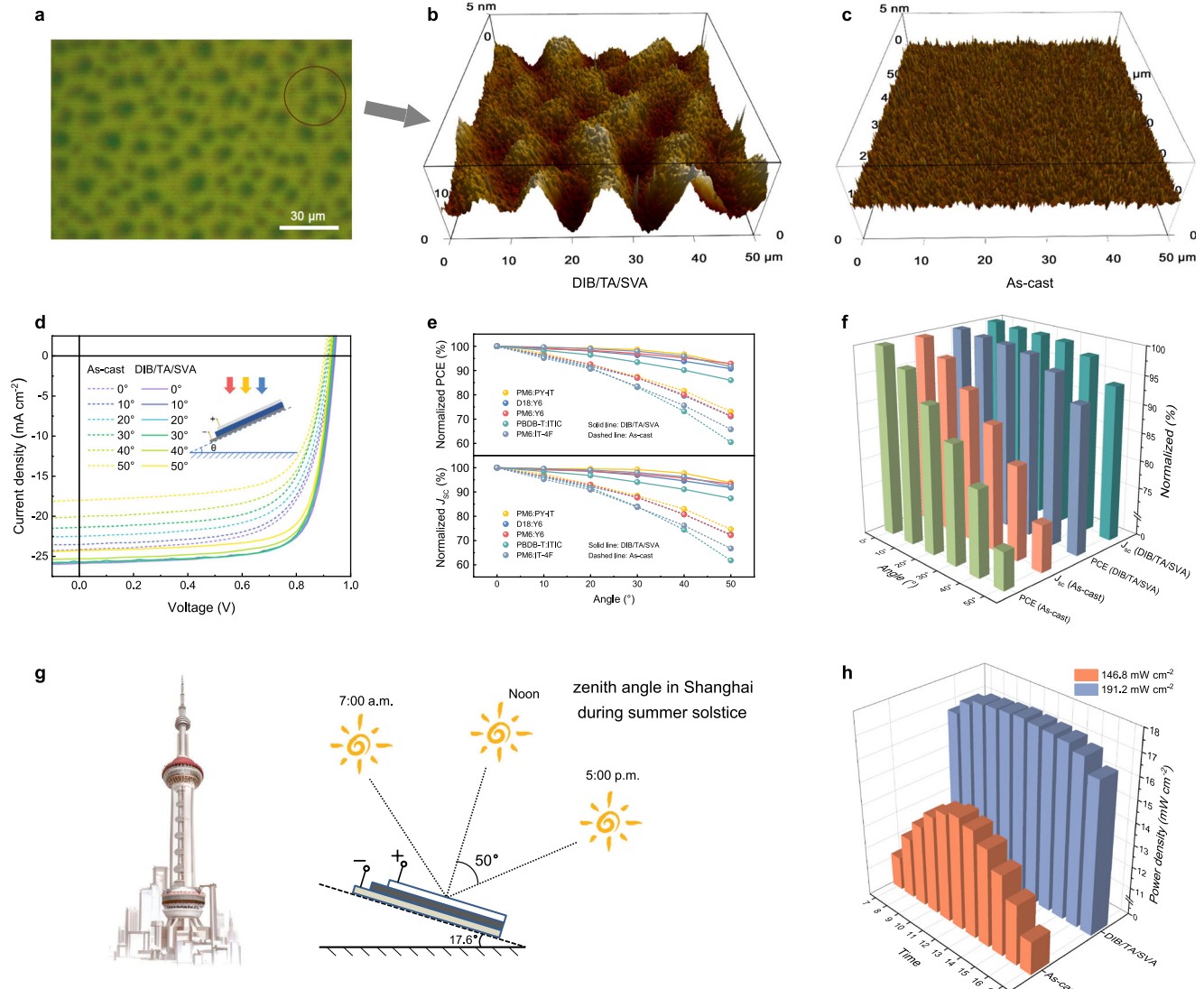

**Fig. 4 | Microscale surface optical morphology characteristics. a** Optical microscope photograph of DIB/TA/SVA. **b** Three-dimensional AFM photograph of DIB/TA/SVA. **c** Three-dimensional AFM photograph of as-cast. **d** *J-V* curve of as-cast and DIB/TA/SVA as function of light receiving angle. **e** Universality test for variable angle experiment. **f** Variation of $J_{SC}$ and PCE of as-cast and DIB/TA/SVA device in different light receiving angle. **g** Scheme for daily power output simulation at the summer solstice in Shanghai. **h** Efficiency corresponding to the incident angle at different times in as-cast and DIB/TA/SVA condition.

that DIB/TA/SVA processing leads to a more favorable donor-acceptor interaction in mixing region, thus hole-transfer process is expedited. The decreased $\tau_2$ upon enhanced crystallization suggests an enhanced diffusion constant, which is associated with better crystalline packing in PY-IT fibrils. As shown in Fig. 5b, the DIB/TA/SVA blends show the optimal combination, which is characterized by longer exciton and carrier diffusion length, and faster exciton splitting process. The peak at 780 nm in TAS is identified to be the positive polaron signal[38], and its recombination kinetics are shown in Supplementary Fig. 35. The fitted decay lifetimes ($\tau_p$) are 1499, 1955 and 2152 ps for the as-cast, DIB, DIB/TA/SVA processed blends, respectively (Supplementary Table 13). Thus, the carrier recombination is much lower in DIB/TA/SVA blends, which agrees well with the photocurrent density-effective voltage ($J_{ph}$-$V_{eff}$) characterization results (Supplementary Fig. 36). The carrier collection efficiency $P(E, T)$ values are 96.09%, 97.20%, and 97.85%, for the as-cast, DIB, and DIB/TA/SVA blends, respectively, contributing to high $J_{SC}$ for corresponding devices.

The charge transport and recombination were investigated by transient photovoltage (TPV) and transient photocurrent (TPC) measurements[39]. The fitted charge lifetimes ($\tau_c$) and charge density ($n$)

under different $V_{OC}$ conditions are summarized in Supplementary Fig. 37. Both $\tau_c$ and $n$ for DIB/TA/SVA devices show higher value than those in other devices over the applied $V_{OC}$ region, which is ascribed to a reduced charge recombination, suggesting less defects and energetic disorder in DIB/TA/SVA blends. The charge lifetime as a function of charge density is shown in Fig. 5c, which follows an approximately exponential law of

$$\tau_c = \tau_0 \left( \frac{n}{n_0} \right)^\lambda \tag{1}$$

where $\tau_0$ and $n_0$ are constants, and the exponential factor $\lambda$ is in relation to the non-geminate recombination order $R$ ($R = \lambda + 1$). When bimolecular recombination is the dominant loss mechanism in device, $R$ is approaching to be 2. When trap-assisted recombination is dominant, $R$ is expected to be higher than 2. The DIB/TA/SVA device shows a minimum $R$ of 2.30 with the maximum lifetimes, which indicates minimized non-geminate recombination. Similar conclusions can be drawn from light intensity ($P_{light}$)-dependent $J_{SC}$ and $V_{OC}$ measurements (Supplementary Fig. 38 and Supplementary Table 14).

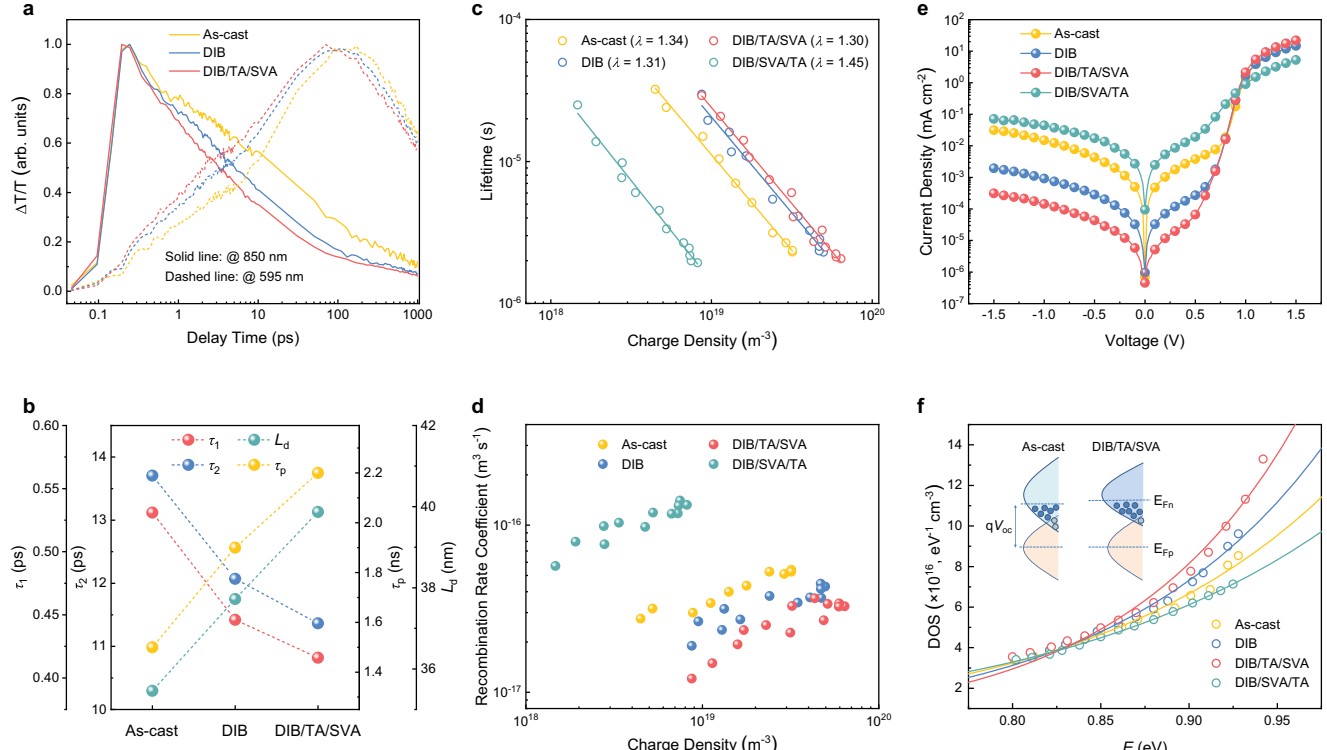

**Fig. 5 | Ultrafast process and devices physical characteristics. a** Hole transfer process kinetics in corresponding conditions. **b** Charge-transfer time and lifetime achieved through multi-exponential fitting in corresponding conditions. ($\tau_1$ is the fast components represent the kinetics of the exciton dissociation in mixing domain or at interfaces; $\tau_2$ is the slow components represent the kinetics of exciton diffusion in crystalline domain to interfaces; $L_d$ is the exciton diffusion lengths; $\tau_p$ is the fitted decay lifetimes for positive polaron signal). **c** Derived charge lifetime as a function of charge density fitted from TPV and TPC results ($\lambda$ is the exponential factor). **d** Recombination rate coefficient as a function of charge density fitted from TPV and TPC results. **e** Dark *J-V* curves for corresponding conditions. **f** Derived LUMO DoS from the capacitance spectra of devices, inset is the schematic illustration.

The non-geminate recombination rate coefficient $k(n)$ follows the equation of

$$k(n) = \frac{1}{\tau_c(n) \times n} \quad (2)$$

where $\tau_c(n)$ is the carrier lifetime. The $k(n)$ with different charge densities is shown in Fig. 5d[40]. A clearly suppressed recombination is seen for DIB/TA/SVA blended thin film over a wide range of charge densities. The space charge limits current (SCLC) measurement shown in Supplementary Table 15 and Supplementary Fig. 39 also supports this result.

The dark *J-V* curves of devices are shown in Fig. 5e. The DIB/TA/SVA device shows reduced reverse saturated current densities, indicating improved diode characteristics with a low leaking current. In the forward direction, the DIB/TA/SVA device shows an early transition from Ohmic to semiconductor zones and higher saturation value, indicating a higher charge density[41,42]. The calculated rectification ratios ($f_{rec}$) at ±1.0 V are $4.8 \times 10^2$, $7.6 \times 10^3$, and $7.1 \times 10^4$ for as-cast, DIB, and DIB/TA/SVA devices, respectively, which is in consistence with the FFs[43]. The fundamental electronic property of the blended thin films is studied by the electron density of state (DoS) characterization, following the relationship of

$$g_n(E) = \frac{N_t}{E_t} \exp\left[-\frac{E - E_{LUMO}}{E_t}\right] \quad (3)$$

where $N_t$ is the total density per unit volume, $E_{LUMO}$ is the LUMO energy level, and $E_t$ is the energy for exponential tail distribution that describes energetic disorder[44]. As shown in Supplementary Table 16

and Fig. 5f, the as-cast device shows a $N_t$ of $3.36 \times 10^{18}$ cm$^{-3}$ eV$^{-1}$ and an $E_t$ of 139 meV. The DIB/TA/SVA device shows a much larger $N_t$ of $1.28 \times 10^{19}$ cm$^{-3}$ eV$^{-1}$ and a smaller $E_t$ of 99 meV, indicating a higher and narrower DoS distribution (inset in Fig. 5f). The device $N_t$ agrees well with the $n$ value obtained in TPC/TPV measurement, validating that the increased charge density is the major contribution to the $J_{SC}$ improvement in DIB/TA/SVA devices. The enhanced DoS and reduced energetic disorder also help to reduce the non-radiative energy loss, as evidenced by the enhanced electroluminescence quantum yields (Supplementary Fig. 40). All characterization details are summarized in Supplementary Methods.

## Discussion

The above-discussed results fully visualize the structure-property relationship of nano-to-micron hierarchical morphology on device performances. The intergradation of bicontinuous network on an island-like rugged surface topology yield unprecedented device performance and power output. To gain a panoramic view of the internal correlations of many factors, a multi-variable cross-correlation analysis is carried out using multiple parameters obtained from many independent observations (Supplementary Fig. 41). The correlation coefficients ($r$) represent the correlation strength. It is seen the area of lamellar and π-π stacking have a favorable $r$ value with $\mu_e$, $\mu_h$, $\alpha$ and $s$, since the fibrillar morphology constructs efficient pathways for charge transport and suppresses recombination. The transport and recombination characteristics show a high $r$-value with $J_{SC}$ and FF. The main factor that determines $J_{SC}$ is the exciton dissociation process ($\tau_1$ and $\tau_2$), and high $r$-values between $\tau_1/\tau_2$ and $J_{SC}$ are recorded. The presence of surface pattern also plays a positive role in the increase of $J_{SC}$. Likewise, surface pattern also contributes greatly on light-receiving

angle tolerance. The significantly improved FF in DIB/TA/SVA results from high mobility, high carrier density, and less recombination. The $V_{OC}$ also gets increased for hierarchical morphology devices due to better thin film electronic structure and device transport characters.

To conclude, a nano-to-micron hierarchical active layer morphology is achieved by using the phase transformative additive and sequential thin film processing strategies for all-polymer organic solar cells. As a result, the optimized nano-sized donor-acceptor fibrillar network integrated in micron-sized island-like rugged thin film surface can better trap the light radiation in the active layer, expedite the exciton-to-polaron conversion, and eventually lead to highly efficient light-to-electricity conversion and large light-receiving incident angles. The synergy of the two unique features in morphology improve the photon and carrier extraction simultaneously, which yields an unprecedented efficiency of 19.06% with superior illumination stability up to 1200 h. In addition, elementary solar solstice power output estimation yields a ~30% increase for operation due to the benefit of large light-receiving angle, which also helps to reduce the limitation of installation form factors. These advances are highly beneficial for the commercial applications of organic solar cells.

## Methods
### Materials
All reagents and chemicals were purchased from commercial sources (Aldrich or Acros) without further purification. PM6 and PY-IT were purchased from Solarmer Materials Inc.

### General methods
UV-vis absorption spectra were recorded in a 0.1 mm slit width cuvette on a Shimadzu spectrometer model UV-1800 with films on the quartz plates at room temperature. The morphologies of the nanostructures were characterized by transmission electron microscopy (TEM, JEM-1400, JEOL, Japan). The GIWAXS characterization of the thin films was performed at the Advanced Light Source (ALS) on beamline 7.3.3 (Lawrence Berkeley National Laboratory, LBNL). The incidence angle was 0.16°, and the beam energy was 10 keV. Samples were prepared under device conditions on the Si substrates. RSoXS was performed at beamline 11.0.1.2 (ALS, LBNL). Samples were prepared under device conditions on the Si/PEDOT:PSS substrates, then placed in water and transferred to a silicon nitride window.

### Device fabrication
All polymer solar cells with structure of ITO/PEDOT:PSS/PM6:PY-IT/PNDIT-F3N/Ag were fabricated. Patterned ITO glass was successively cleaned twice in an ultrasonic bath by detergent, deionized water, acetone and isopropyl alcohol for 15 min each and then dried under dry oven. The precleaned substrates were treated in an ultraviolet-ozone chamber for 20 min, then a ~15 nm thick PEDOT:PSS (Clevious PVP AI 4083 H. C. Stark, Germany) thin film was deposited onto the ITO surface by spin-coating (4000 rpm) and baked at 150 °C for 20 min. The PEDOT:PSS solution was diluted by water in volume fraction 1:1. Active layer was fabricated by spin-coating mixed solution of PM6 and PY-IT (weight ratio of 1:1.2) in precursor CF solution (with 30 mg mL$^{-1}$ solid additive of DIB) at the donor concentration of 7 mg mL$^{-1}$, at 3500 rpm for 40 s on the PEDOT:PSS substrate in a nitrogen glove box, and then baked for 10 min at 100 °C. Then SVA treatment is done in an opaque jar, and ~200 microliter of carbon disulfide ($CS_2$) is added to serve as swelling vapor in closed environment after ~5 min of SVA treatment, the blended thin film is taken out, and then it is washed by isopropanol (spin-coating at 3500 rpm for 30 s). The active layer (around 120 nm) was pre-coated by methyl alcohol at 3500 rpm for 30 s, and PNDIT-F3N (0.7 mg mL$^{-1}$ in methyl alcohol with 0.3% acetic acid, v/v) was fabricated at 2600 rpm for 30 s on BHJ film. Finally, 120 nm thick silver layer was thermally evaporated at a base pressure of $1 \times 10^{-6}$ mbar in an evaporate chamber. The evaporation thickness was controlled by SQC-310C deposition controller (INFICON, Germany). The monomolecular layer of 2PACz as HTL is prepared was fabricated by spin-coating 0.3 mg mL$^{-1}$ methanol solution at 3000 rpm for 30 s on the ITO substrate in a nitrogen glove box and baked for 6 min at 100 °C, then flushed by methanol at 4000 rpm for 20 s.

### Device characterization
The $J-V$ curves were measured with Keithley 2400 Source under the illumination of AM 1.5 G irradiation (100 mW cm$^{-2}$) using a 150 W solar simulator (DM-40S3, SAN-EI ELECTRIC, Japan) in glove box at room temperature. The $J-V$ characteristics were measured along the forward scan direction from −0.2 to 1 V, with a scan step of 50 mV and a dwell time is 10 ms using a Keithley 2400 Source Measure Unit. EQE spectra were measured by using a solar-cell spectral-response measurement system (QE-R3011, Enlitech).

### Reporting summary
Further information on research design is available in the Nature Portfolio Reporting Summary linked to this article.

## Data availability
The data that support the plots within this paper and other findings of this study are available from the corresponding authors upon request. The raw data has been uploaded to the database. Source data are provided with this paper in the Materials Cloud database (https://doi.org/10.24435/materialscloud:g4-jw). Source data are provided with this paper.

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

## Acknowledgements

This work was financially supported by National Key R&D Program of China (grant no. 2021YFB4001302), the National Natural Science Foundation of China (grant nos. 51973110 and 22109094), the Science and Technology Commission of Shanghai Municipality (grant nos. 20ZR1426200, 21DZ1208600), the Key research project of Shandong Province (grant no. 2020CXGC010403), the Center of Hydrogen Science, Shanghai Jiao Tong University, China. We thank Cheng Wang and Chenhui Zhu from Advanced Light Source for providing X-ray scattering tests, which were carried out at beamline 7.3.3 and 11.0.1.2 at the Advanced Light Source, Molecular Foundry, Lawrence Berkeley National Laboratory, supported by the DOE, Office of Science, and Office of Basic Energy Sciences. Le.Z. acknowledge the funding supported by China Postdoctoral Science Foundation (Grant No. 2022T150406). M.Z. acknowledge the funding supported by Postdoctoral Innovative Talent Support Program (Grant No. BX20220203) and China Postdoctoral Science Foundation (Grant No. 2022M722072).

## Author contributions

F.L. conceived and directed this project. R.Z. fabricated and characterized the OPV devices. R.Z. and Le.Z. conducted the certification. M.Z. Ye.Z. and T.H. carried out the TPV, TPC, impedance characterization and analyzed the data. M.Z. and N.H. supplied AFM-IR results and corresponding analysis. W.Z. carried out the GIWAXS and RSoXS measurement and assisted with data analysis. G.Z. and H.Z. provided TA results and corresponding analysis. J.Z. conducted the AFM measurements. Z.Z. conducted the TEM measurements. F.H. and Y.B. contributed to the simulation results. H.W. and Z.T. carried out the energy loss measurements. Li.Z., X.X., H. J. and Yo.Z. contributed to the fruitful discussion of this project. R.Z. wrote the manuscript, and Le.Z., M.Z., W.Z., T.H., C.-C.C. and F.L. contributed to revisions of the manuscript. This manuscript was mainly prepared by F.L., R.Z., Le.Z., and all authors participated in the manuscript preparation and commented on the manuscript.

## Competing interests

The authors declare no competing interests.
