## [Peer Review File · Nature Communications]

All-polymer organic solar cells with nano-to-micron hierarchical morphology and large light receiving angleEditorial Note: This manuscript has been previously reviewed at another journal that is not operating a transparent peer review scheme. This document only contains reviewer comments and rebuttal letters for versions considered at *Nature Communications*. Mentions of the other journal have been redacted.

REVIEWER COMMENTS

Reviewer #1 (Remarks to the Author):

I appreciate the detailed and thorough responses that the authors provided to all reviewer comments. I believe this has significantly improved the manuscript, which already presented impactful results, and in my opinion the present version is ready for publication.

Reviewer #2 (Remarks to the Author):

This manuscript demonstrated the high efficiency record of the all polymer PV devices through engineering improvement. The classic polymer engineering methods, such like additive control and thermal and solvent annealing treatments, were systematically investigated to engineer the structure and morphology of the devices and evaluate the impacts. Comparing to the first submission, the rebuttal has included even more characterizations, such like optical images and stability of I-V curves, and better explanations on the results, such like the EQE calculation and the exploration of the processing mechanism. However, there are some over-explanations in the discussion. One example is the 'g-factor', the paracrystalline disordered factor. The calculation is based on the coupling of d-spacing and the FWHM of the pi-stacking. As the d-spacing of the pi stacking are the same for all treated samples, the g-factor is almost the same to FWHM, correlated to the grain size (CCL in manuscript). The g-factor is unnecessary here. Meanwhile, the calculation of area of the scattering peaks is meaningless as the inconsistency of the samples.

Overall, the manuscript is much better presented. The new evidences are convincing and the language is much better. I recommend for publication if the minor changes are addressed.

Reviewer #3 (Remarks to the Author):

This manuscript is a re-review of a submission that I received from [REDACTED] in which I and another Reviewer (1) raised a number of major concerns regarding consistency and interpretation. There were also questions as to the novelty of the findings in light of previous works – notably a Nature Materials publication which was omitted from the referencing. I believe this new manuscript is a significant improvement in two aspects: i) a further enhancement in device performance using the hole transport

layer 2PACZ – a certified PCE which is indeed the highest for a polymer-polymer blend; and ii) a considerable body of new measurements and simulations. From my initial concerns, I would note the following specific responses:

1. The energetics of the PY-IT and blends were inconsistent as originally determined by UPS/IPES relative to the EQE edge, PL and absorption. Additional CV measurements have somewhat clarified these inconsistencies – I take the point that electrical and optical gaps are quite often different, but the initial large inconsistencies should have been investigated in the original submission. The new analysis provides a level of certainty that is acceptable within the normal bounds of OPV materials.
2. The high Voc was not properly justified and investigated in the original manuscript. With clarification of the optical gaps and a more complete analysis via reciprocity and the radiative and non-radiative voltage losses as suggested, the 0.94 Voc looks realistic.
3. The EQE shapes have been qualified by additional electro-optical simulations and the addition of the IQE determination is useful to show the relatively balanced efficiency of Channel I and Channel II processes for electron and hole transfer respectively.
4. Errors in stated mask areas have been clarified and corrected.
5. The Specific Responsivity certification measurement has been qualified and re-checked by comparison of the lab EQE with the certification EQE. This was absolutely crucial to validate the certification. I would recommend Figure R16 be included in the SI at the very least and a suitable explanation provided as to how this validates the SR. I note, this revised manuscript contains the certification for the new 2PACZ devices – and not an EQE comparative analysis of the original devices from the [REDACTED] submission. I further note that the new certification is from the Chengdu Institute of Product Quality Inspection, whilst the original certification was obtained from the Fujian Metrology Institute – and no Specific Responsivity measurement is included for the new device certification.
6. Several minor inconsistencies and errors have now been rectified but the manuscript is still unclear in parts and requires a thorough copy edit.

Thus, I would recommend publication but only subject to the following:

1. Figure R16 be inserted with suitable explanations of the EQE comparison.
2. Specific Responsivity be included as part of the certification and not just the IV (Page 2 of the Test Report). I believe that readers should be able to do the comparative analysis EQE vs. SR as a matter of course in certified record device reports, and I would note that the raw data will be made available upon reasonable request.
3. The manuscript receives a thorough copy edit.

Responses to the reviewers' reports for NENERGY-22050877

(Text in blue is our responses)

Reviewer #1 (Remarks to the Author):

I appreciate the detailed and thorough responses that the authors provided to all reviewer comments. I believe this has significantly improved the manuscript, which already presented impactful results, and in my opinion the present version is ready for publication.

Response:

We are very grateful to the reviewer for supporting this research published in the journal of *Nature Communications*.

Reviewer #2 (Remarks to the Author):

This manuscript demonstrated the high efficiency record of the all polymer PV devices through engineering improvement. The classic polymer engineering methods, such like additive control and thermal and solvent annealing treatments, were systematically investigated to engineer the structure and morphology of the devices and evaluate the impacts. Comparing to the first submission, the rebuttal has included even more characterizations, such like optical images and stability of I-V curves, and better explanations on the results, such like the EQE calculation and the exploration of the processing mechanism. However, there are some over-explanations in the discussion. One example is the 'g-factor', the paracrystalline disordered factor. The calculation is based on the coupling of d-spacing and the FWHM of the pi-stacking. As the d-spacing of the pi stacking are the same for all treated samples, the g-factor is almost the same to FWHM, correlated to the grain size (CCL in manuscript). The g-factor is unnecessary here. Meanwhile, the calculation of area of the scattering peaks is meaningless as the inconsistency of the samples.

Overall, the manuscript is much better presented. The new evidences are convincing and the language is much better. I recommend for publication if the minor changes are addressed.

Response:

Thanks for the comment. We have removed the excessive description of the 'g-factor' in the revised version. The scattering peak area is used to qualitatively analyse the crystallinity of the samples. We prepared samples with the same film thickness (120 nm), using the same size wafer substrate, and completed the tests using the same conditions in the same experiment, so that the scattering peak area can be used for qualitative analysis of the crystallinity of the sample.

Reviewer #3 (Remarks to the Author):

This manuscript is a re-review of a submission that I received from [REDACTED] in which I and another Reviewer (1) raised a number of major concerns regarding consistency and interpretation. There were also questions as to the novelty of the findings in light of previous works – notably a Nature Materials publication which was omitted from the referencing. I believe this new

manuscript is a significant improvement in two aspects: i) a further enhancement in device performance using the hole transport layer 2PACZ; a certified PCE which is indeed the highest for a polymer-polymer blend; and ii) a considerable body of new measurements and simulations. From my initial concerns, I would note the following specific responses:

1. The energetics of the PY-IT and blends were inconsistent as originally determined by UPS/IPES relative to the EQE edge, PL and absorption. Additional CV measurements have somewhat clarified these inconsistencies; I take the point that electrical and optical gaps are quite often different, but the initial large inconsistencies should have been investigated in the original submission. The new analysis provides a level of certainty that is acceptable within the normal bounds of OPV materials.
2. The high V_{oc} was not properly justified and investigated in the original manuscript. With clarification of the optical gaps and a more complete analysis via reciprocity and the radiative and non-radiative voltage losses as suggested, the 0.94 V_{oc} looks realistic.
3. The EQE shapes have been qualified by additional electro-optical simulations and the addition of the IQE determination is useful to show the relatively balanced efficiency of Channel I and Channel II processes for electron and hole transfer respectively.
4. Errors in stated mask areas have been clarified and corrected.
5. The Specific Responsivity certification measurement has been qualified and re-checked by comparison of the lab EQE with the certification EQE. This was absolutely crucial to validate the certification. I would recommend Figure R16 be included in the SI at the very least and a suitable explanation provided as to how this validates the SR. I note, this revised manuscript contains the certification for the new 2PACZ devices; and not an EQE comparative analysis of the original devices from the [REDACTED] submission. I further note that the new certification is from the Chengdu Institute of Product Quality Inspection, whilst the original certification was obtained from the Fujian Metrology Institute; and no Specific Responsivity measurement is included for the new device certification.
6. Several minor inconsistencies and errors have now been rectified but the manuscript is still unclear in parts and requires a thorough copy edit.

Thus, I would recommend publication but only subject to the following:

Comment 1:

Figure R16 be inserted with suitable explanations of the EQE comparison.

Response:

Thanks for the comment. The latest certification was carried out at National Photovoltaic Product Quality Inspection & Testing Center (NPPQITC) in Chengdu where is the institution that we can obtain the appropriate testing time. However, NPPQITC does not perform EQE measurement based on their testing procedure. Thus we cannot provide the EQE result in certification report. NPPQITC is an accredited institution that perform national PV quality inspection, and we cannot question their professioncy in measurement procedure and accuracy. Our device afforded a V_{oc} of 0.945 V and J_{sc} of 26.3 mA cm⁻², which is exceptional in OPV devices. We have carried out J-V and EQE test in home lab for numerous of times, and the calculated current is acceptable comparing to J-V results. Thus we are confident about the device performance results.

Comment 2:

Specific Responsivity be included as part of the certification and not just the IV (Page 2 of the Test Report). I believe that readers should be able to do the comparative analysis EQE vs. SR as a matter of course in certified record device reports, and I would note that the raw data will be made available upon reasonable request.

Response:

Thanks for the comments. We take the suggestion that can provide the raw data upon request. During the study, we used the different certification agencies, since that making a testing appointment can be difficult. And the agencies that we used are all qualified and the corresponding testing standards are given in the report. The Chengdu Institute of Product Quality Inspection testing center does not measure EQE. We mailed the device to them, and they did the test and format the report. We provided all the information in our manuscript. These reports are also available to the readers.

Comment 3:

The manuscript receives a thorough copy edit.

Response:

Thanks for the comment. We appreciate the reviewer's effort in pointing out language issues. We have edited the manuscript with the help from the native English speaker.